# Topic-Informed Dialogue Summarization using Topic Distribution and Prompt-based Modeling

**Jaeah You[1]** and **Youngjoong Ko[2]** [*]
[1]Department of Artificial Intelligence
[2]Department of Computer Science and Engineering
Sungkyunkwan University
youjaeah@gmail.com, yjko@skku.edu

## Abstract

Dealing with multiple topics should be considered an important issue in dialogue summarization, because dialogues, unlike documents, are prone to topic drift. Thus, we propose a new dialogue summarization model that reflects dialogue topic distribution to consider all topics present in the dialogue. First, the distribution of dialogue topics is estimated by an effective topic discovery model. Then topic-informed prompt transfers estimated topic distribution information to the output of encoder and decoder vectors. Finally, the topic extractor estimates the summary topic distribution from the output context vector of decoder to distinguish its difference from the dialogue topic distribution. To consider the proportion of each topic distribution appeared in the dialogue, the extractor is trained to reduce the difference between the distributions of the dialogue and the summary. The experimental results on SAMSum and DialogSum show that our model outperforms state-of-the-art methods on ROUGE scores. The human evaluation results also show that our framework well generates comprehensive summaries.

## 1 Introduction

In general, text summarization aims to generate a summary by capturing the core meaning from an original document consistently written by one participant on a single topic, such as news, scientific publications, etc (Rush et al., 2015; Nallapati et al., 2016). On the other hand, since a dialogue consists of multi-speakers, the topic of the dialogue may be changed as a topic drift according to the speaker's intentions (Zhao et al., 2020; Feng et al., 2021a). Therefore, the dialogue summarization should take into account the distribution of multiple topics in a dialogue and reflect this distribution in generating the summary(Zou et al., 2021a).

--------
[*] Corresponding author

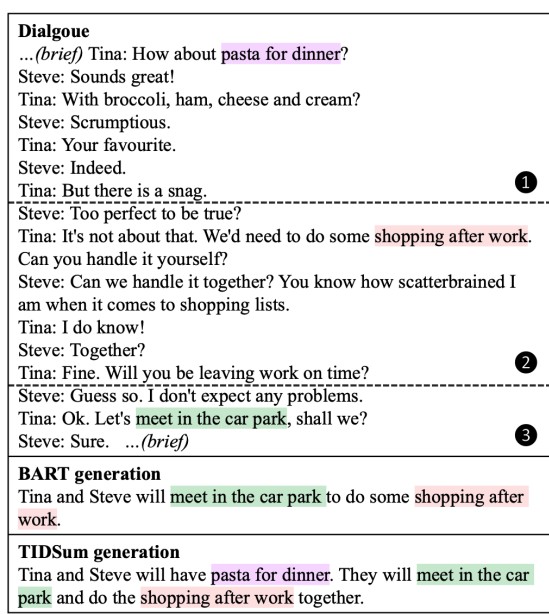

Figure 1: Dialogue summary examples of SAMSum generated by BART and TIDSum.

Figure 1 shows an example of dialogue summary generation results from BART (Lewis et al., 2019) and TIDSum models in the SAMSum dataset (Gliwa et al., 2019); BART is a baseline model that has been widely used due to high performance in summary tasks. The content of the example dialogue in Figure 1 can be divided into three parts: ❶ Tina and Steve are having pasta for dinner, ❷ they will do the shopping together, and ❸ they make an appointment to meet in the car park after Steve finishes work. That is, we can think that the dialogue contains three topics. However, BART did not capture the most important purpose of Tina and Steve's appointment to have pasta for dinner. They made an appointment to go shopping for ingredients for pasta in dinner, so this should be included in the summary. Therefore, we focus on generating a more comprehensive summary that captures all the topics in the dialogue, without missing an important topic.

In this paper, we propose a novel model, **T**opic-**I**nformed **D**ialogue **Sum**marizer (**TIDSum**), that

generates a summary by considering the distribution of topics existing in the dialogue. To estimate the distribution of multiple topics in a dialogue, we exploit the *TopClus* model to obtain the distribution for the input dialogue, which was developed for automatic topic discovery from text corpora (Meng et al., 2022). Moreover, a task-specific soft-prompt, namely *the topic-informed prompt*, is added to make the encoder context vector[1] well capture the dialogue topic information in the decoding phase as well as the encoding phase. The topic-informed prompt is created by the latent embedding of the auto-encoder in the *TopClus* model. Since the latent embedding of dialogue from the *TopClus* model sufficiently contains the topic information of the dialogue, the topic-informed prompt can influence the context vector of other tokens in the encoder and decoder of summarizer through the encoder self-attention and decoder cross-attention processes. The output context vectors from the decoder are used for the topic extractor to estimate the topic distribution of the generated summary with it. Then the topic extractor provides an auxiliary loss function to reduce the difference between the dialogue topic distribution and the summary topic distribution in the training phase. This learning approach generates a better summary that well reflects the topic of the dialogue.

In the experiments, two daily dialogue summarization datasets, *SAMSum* and *DialogSum*, were used to evaluate our model. Compared to the previous model, the proposed model improved the SOTA performance by 1.19%p and 1.94%p in Rouge-1 for *SAMSum* and *DialogSum*, respectively.

## 2 Related Work

Recently, there has been increasing attention on neural summarization for dialogues. Current studies mainly apply transformer-based models (e.g., BART (Lewis et al., 2019)) to abstractly summarize dialogues. However, these models are pre-trained on generic text corpora and it is essential to fine-tune them in a specific way for dialog data. Many studies have investigated how to find topics in dialogues. Zhao et al. (2020) modeled the dialogue as an interactive graph according to the topic word information extracted from LDA (Blei et al., 2003). Feng et al. (2021b), which used DialoGPT (Zhang et al., 2019) as the annotator, performed three dia-

logue annotation tasks, including keywords extraction, redundancy detection, and topic segmentation. Liu et al. (2021) proposed two topic-aware contrastive learning objectives. This method implicitly modeled the topic change and handled information scattering challenges for the dialogue summarization. Since summarizing dialogues is essential in customer services, Zou et al. (2021b) proposed a topic-augmented two-stage summarizer with a multi-role-specific topic modeling mechanism. Li et al. (2022) presented a novel curriculum-based prompt learning and applied a topic-aware prompt, from which we got the idea for a topic-informed prompt.

## 3 Topic-informed Summary Generation Framework

To perform the dialogue summarization effectively, it is necessary to identify the distribution of topics in the dialogue scattered across multiple utterances. Therefore, we propose a model to generate a topic-informed summary by reflecting the dialogue topic distribution to the summary topic distribution. The base architecture of TIDSum is a Transformer-based auto-regressive language model, BART.

### 3.1 Topic-Informed Prompt

In Figure 2-(1), we input a dialogue into *TopClus* to obtain the latent topic embedding and dialogue topic distribution. To be specific, the **l**atent **t**opic **e**mbedding $lte$ is derived from the auto-encoder structure, *TopClus*, which ignores extraneous elements and contains only salient information from the input. Each topic $t_k$ is associated with the dialogue-topic distribution $p(t_k|lte)$, where $k$ is the number of topics[2]. The distribution not only represents all the topical information present in the dialogue but also distinguishes between important and unimportant topics. As you can see in Figure 2-(1), the **t**opic-**i**nformed **p**rompt $tip$ is created by concatenating two $lte$s to match the dimensions of BART. Since $lte$ is a hidden state of the *TopClus*, an auto-encoder structure, it must be smaller than the input dimension of *TopClus*. The *TopClus* input is fixed at 768 dimensions because it origins from the CLS token of BERT that encodes the input dialogue. Thus, we set it to 512 dimensions to easily match the input dimensions of BART-large (1024 dims) by concatenating two

---

[1]The last hidden states of the encoder that are used as the decoder input are called the encoder context vector.

[2]We set the number of topics to 5 for SAMSum and 7 for DialogSum, which show the highest performance.

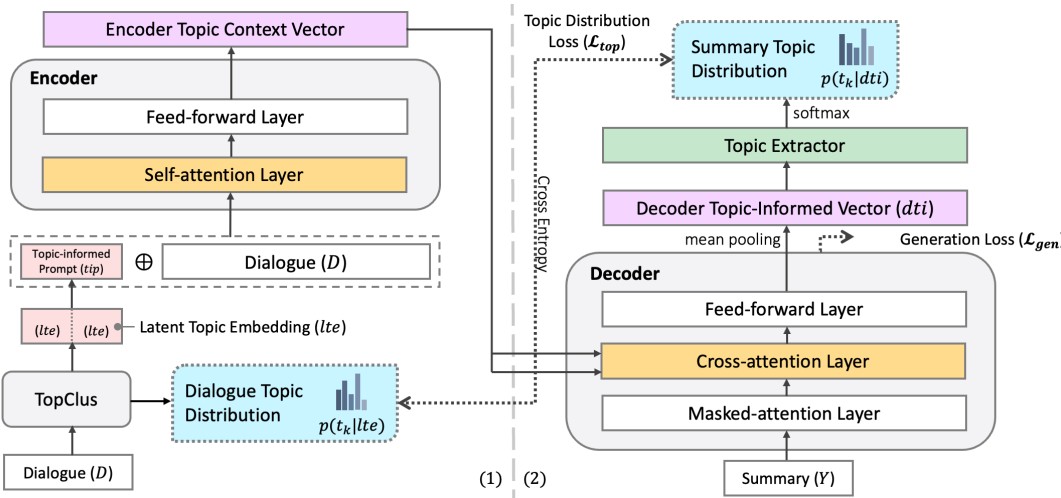

Figure 2: Overall framework of the proposed dialogue summarizer. (1)-(2) represent the order in which TIDSum operates.

$lte$s (1024 dims). Then, $tip$ is located at the first token of the encoder input, preceding the dialogue $\mathcal{D} = \{w_1, w_2, ..., w_n\}$ with $n$ tokens. The final input form is $\mathcal{X} = \{tip, w_1, w_2, ..., w_n\}$. The topic information of the prompt $tip$ infuses each token through a process known as encoder self-attention during training. In this manner, the encoder context vectors are topically enhanced within the encoder and their topic information is also propagated into the tokens of the decoder via encoder-decoder cross-attention.

### 3.2 Topic Extractor

In Figure 2-(2), the topic extractor, composed of MLP, serves to extract the distribution of the summary. Through encoder-decoder attention, the topic information sourced from $tip$ is reflected in the decoder tokens. Therefore, we perform mean pooling on all tokens of the decoder to get the decoder topic-informed vector $dti$ as follows:

$$dti = \frac{1}{M} \sum_{m=1}^{M} y_m \qquad (1)$$

where $M$ is a length of the summary and $\mathcal{Y} = \{y_1, y_2, ..., y_m\}$ is the corresponding summary of $m$ tokens. The topic extractor estimates a summary topic distribution $p(t_k|dti)$ of $k$ topics from $dti$. It is trained with a cross-entropy loss to reduce the difference between the dialogue topic distribution and the summary topic distribution. The topic distribution loss $\mathcal{L}_{top}$ is formulated as follows:

$$\mathcal{L}_{top} = -\sum_{k=1}^{K} p(t_k|lte) \log p(t_k|dti) \qquad (2)$$

where $K$ is the number of topics. This ensures that the summary $\mathcal{Y}$ is generated to reflect the dialogue topics.

### 3.3 Topic-informed Summary Generation

The generation loss $\mathcal{L}_{gen}$ is typically defined as the negative log-likelihood of the target summary given the input dialogue.

$$\mathcal{L}_{gen} = -\sum_{m=1}^{M} \log p(y_m|y_{1:m-1}, \mathcal{X}) \qquad (3)$$

The final loss $\mathcal{L}_{final}$ is a weighted sum of the generation loss $\mathcal{L}_{gen}$ and the topic distribution loss $\mathcal{L}_{top}$:

$$\mathcal{L}_{final} = \mathcal{L}_{gen} + \lambda \mathcal{L}_{top} \qquad (4)$$

where $\lambda$ is a hyperparameter that controls the relative importance of these two losses. Experimentally, setting $\lambda$ to 0.75 showed the best performance. By minimizing the final loss $\mathcal{L}_{final}$ during training, the model is encouraged to generate summaries that are faithful to the input dialogue and well reflect the topic distribution of the dialogue.

## 4 Experiments

### 4.1 Datasets

Two dialogue summarization datasets, SAMSum (Gliwa et al., 2019) and DialogSum (Chen et al., 2021), were used to verify the proposed model. SAMSum is an online chit-chat dataset from WhatsApp and WeChat. DialogSum is a real-life dialogue dataset containing diverse task-oriented scenarios and topics. It consists of a formal style of dialogue. Table 1 shows the additional details for each dataset.

| Dataset | SAMSum | DialogSum |
|---|---|---|
| Language style | Written | Spoken |
| Domain | Online Chat | Daily life Dialogue |
| Train/Valid/Test | 14,732/818/819 | 12,460/500/500 |
| Avg. speakers | 2.2 | 2 |
| Avg. turns | 8.4 | 13.8 |
| Avg. dia/sum length | 94/25 | 131/23 |

Table 1: Data description for SAMSum and DialogSum.

| Model | R-1 | R-2 | R-L |
|---|---|---|---|
| **SAMSum** | | | |
| Transformer (Vaswani et al., 2017) | 36.62 | 11.18 | 33.06 |
| BART-Large (Lewis et al., 2019) | 53.06 | 28.18 | 49.50 |
| TGDGA (Zhao et al., 2020) | 43.11 | 19.15 | 40.49 |
| BART($\mathcal{D}_{ALL}$) (Feng et al., 2021b) | 53.70 | 28.79 | 50.81 |
| Ctrl-DiaSumm+Coref+DA (Liu and Chen, 2021) | 56.0 | 31.7 | 54.1 |
| ReWriteSum (Fang et al., 2022) | 54.20 | 27.10 | 50.10 |
| ConFiT (Tang et al., 2022) | 53.89 | 28.85 | 49.29 |
| SICK++ (Kim et al., 2022) | 53.24 | 28.10 | 48.90 |
| (Li et al., 2022) | 55.97 | 31.67 | 52.32 |
| *Our Architecture* | | | |
| TIDSum | **57.19** | **33.41** | **55.13** |
| w\o Topic-informed prompt | 56.77 | 32.90 | 54.99 |
| w\o Topic extractor | 56.62 | 32.31 | 54.92 |
| **DialogSum** | | | |
| Transformer (Vaswani et al., 2017) | 35.91 | 8.74 | 33.50 |
| BART-Large (Lewis et al., 2019) | 45.19 | 19.69 | 43.08 |
| ReWriteSum (Fang et al., 2022) | 35.1 | 14.6 | 32.1 |
| SICK++ (Kim et al., 2022) | 46.26 | 20.95 | 41.05 |
| *Our Architecture* | | | |
| TIDSum | **48.20** | **21.80** | **46.15** |
| w\o Topic-informed prompt | 47.52 | 21.43 | 45.16 |
| w\o Topic extractor | 47.26 | 21.27 | 45.05 |

Table 2: ROUGE scores on the SAMSum from baseline models and proposed methods

## 4.2 Experimental Settings

We loaded the pre-trained "facebook/bart-large"[3] for initialization. The learning rates of SAMSum and DialogSum were set to 1e-5 and 3e-5, and the train batch size were 2 and 4, respectively. The training was conducted at Nvidia Quadro RTX 8000 48G. We employed Py-rouge package to evaluate the models following (Feng et al., 2021b; Liu and Chen, 2021).

## 4.3 Comparison Models

**TGDGA** uses topic information and interactive graph structures. **BART** ($\mathcal{D}_{ALL}$) performs three dialogue annotation tasks using PLM. **ReWrite-Sum** used the utterance rewriting mechanism to complete the omitted content. **ConFiT** is also trained via a novel contrastive fine-tuning. **SICK++** summarized the dialogue in a way that utilizes

[3]https://huggingface.co/facebook/bart-large

commensense knowledge. Li et al. (2022) applies a curriculum-based prompt learning. **Ctrl-DiaSumm+Coref+DA** generates controllable summaries with personal named entity.

## 4.4 Main Results

To evaluate our model, we employed the ROUGE scores, which are widely used in summarization tasks. In detail, the Rouge-1, Rouge-2 and Rouge-L variants, which consider unigram, bigram, and longest common subsequence overlap between generated and reference summaries, were utilized in our experiments (Lin, 2004).

Table 2 provides a comparison of our model with previous approaches on *SAMSum* and *DialogSum*. As shown in Table 2, TIDSum achieved the-state-of-the-art performances on both datasets. TIDSum obtains relative improvements of 1.19%p on Rouge-1, 1.71%p on Rouge-2 and 1.03%p on Rouge-L compared with the previous SOTA model in SAM-Sum, and 1.94%p, 0.85%p and 3.07%p in Dialog-Sum. These results demonstrate the effectiveness of our technique for generating a summary by distinguishing and reflecting on the topics that appear in the dialogue.

In the ablation study, the topic extractor was found to have the most impact on performance, as it is responsible for creating a summary topic distribution in the framework that essentially reflects dialogue topics.

## 4.5 Human Evaluation

| Model | Info. | Conc. | Cov. |
|---|---|---|---|
| Golden summary | 3.83 | 4.19 | 3.70 |
| BART-large | 3.05 | 3.92 | 3.03 |
| TIDSum w\o *tip* | 3.26 | **3.99** | 3.30 |
| TIDSum | **3.69** | 3.97 | **3.71** |

Table 3: Human evaluation on SAMSum. "Info.", "Conc.", and "Cov." stand for Informativeness, Conciseness and coverage, respectively. w\o *tip* means that we do not use the topic-informed prompt.

For qualitative measurement of the generated summaries, we conducted human evaluations on three metrics by just following Feng et al. (2021b). Informativeness evaluates how well the generated summaries capture more salient information. Conciseness measures how well the summary discards redundant information. Coverage measures how well the summary covers each part of the dialogue. We randomly sampled 60 dialogues with

corresponding generated summaries to evaluate the SAMSum dataset. We asked three expert evaluators to rate each metric on a scale of 1 to 5, with higher scores being better. The results are shown in Table 3.

In the case of informativeness, the golden summary has the highest value because it is a summary written by a person. However, TIDSum showed higher performance than BART. Conciseness is probably the shorter, the better, so it was slightly higher for TIDSum using only the topic extractor than for TIDSum infused with more topic information, but the performance was almost the same. Finally, coverage is a metric about whether a summary covers the entire content of the dialog, and TIDSum scored higher than the golden summary on this metric. This result shows that TIDSum effectively covers all the content of the dialogue.

## 5 Analysis

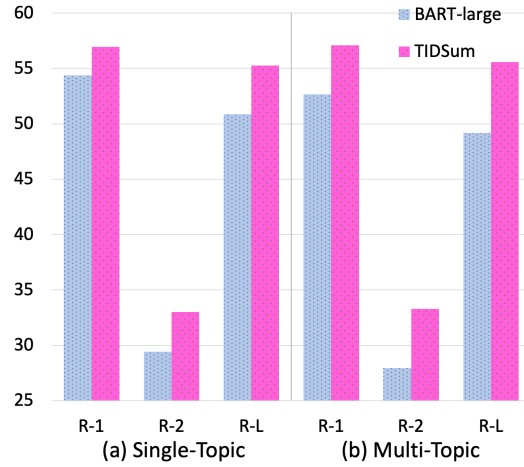

Figure 3: ROUGE scores differences for single-topic and multi-topic dialogues

**Comparison for Single-Topic vs Multi-Topic Dialogues**

We herein attempt to verify that our model works better in multi-topic dialogues than in single-topic dialogues and it can generate comprehensive summaries well. The *SAMSum* test dataset was divided into single-topic and multi-topic ones. Dialogues with an entropy value of topic distribution less than 0.5 were regarded as single-topic ones. Eventually, the total dialogues are separated into 178 single-topic dialogues and 641 multi-topic dialogues. As a result, TIDSum showed larger improvement differences over baseline, BART-large in multi-topic dia-

logues (b) as shown in Figure 3. The result proves that TIDSum is effective for summarizing more multi-topic dialogues. In real-world scenarios, TID-Sum can be applied not only to simple dialogues between two speakers, but also to multi-party dialogues, discussion summarization, etc. with more speakers and various topics. Figure 3 shows that the performance difference between TIDSum and BART-large is much larger in multi-topic than in single-topic. This suggests that it is applicable to dialogues with more diverse topics.

## 6 Conclusion

We propose TIDSum, a novel model for dialogue summarization. By reflecting the distribution of topics in the dialogue, TIDSum generates comprehensive summaries. We utilize the *TopClus* model to estimate topic distributions in the dialogue, and introduce a task-specific soft-prompt, *the topic-informed prompt*, to capture and infuse topic information through the encoding and decoding phases. The generated summaries were evaluated using *SAMSum* and *DialogSum* datasets, and our model outperformed previous approaches with a significant improvement in ROUGE scores and human evaluation results. Overall, TIDSum effectively captures and summarizes the details of each topic in the dialogue, resulting in high-quality summaries.

## Limitations

The proposed method needs to train the *TopClus* model with the dialogue data to get the topic distribution and latent topic embedding of the dialogue before fine-tuning the BART based summarization model. Since *TopClus* is an auto-encoder model with high dimensional layers, it takes a long time to train. With the obtained topic distribution and latent topic embedding, the BART based summarization model is trained and generates a summary in inference phase. This two-stage process is complicated and requires some time. Therefore, in order to simplify this process, our future work is to incorporate only the essential parts of *TopClus* into the main learning process.

## Acknowledgements

This work was supported in part by the National Research Foundation of Korea (NRF) grant funded by the Korea government (MSIT) (No. NRF-2020R1A2C2100362), in part by Institute for Infor-

mation & communications Technology Planning & Evaluation (IITP) grant funded by the Korea government (MSIT) (No. 2022-0-00369, (Part 4) Development of AI Technology to support Expert Decision-making that can Explain the Reasons/Grounds for Judgment Results based on Expert Knowledge), in part by Institute of Information & Communications Technology Planning & Evaluation (IITP) grant funded by the Korea Government (MSIT) (No. 2020-0-00368, A Neural-Symbolic Model for Knowledge Acquisition and Inference Techniques), and in part by Institute of Information & communications Technology Planning & Evaluation (IITP) grant funded by the Korea government (MSIT) (No.2019-0-00421, AI Graduate School Support Program (Sungkyunkwan University))

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
