# OpenReview forum: "Topic-Informed Dialogue Summarization using Topic Distribution and Prompt-based Modeling"
_EMNLP/2023/Conference — EMNLP 2023 Findings_

### Official Review · Reviewer_Fkky · 2023-07-31

**Soundness:** 3

**Excitement:**

3: Ambivalent: It has merits (e.g., it reports state-of-the-art results, the idea is nice), but there are key weaknesses (e.g., it describes incremental work), and it can significantly benefit from another round of revision. However, I won't object to accepting it if my co-reviewers champion it.

**Paper Topic And Main Contributions:**

This paper proposed a new model for Topic-Informed Dialogue Summarization that considers all topics present in a dialogue, reflecting dialogue topic distribution. The model estimates the distribution of dialogue topics and generates comprehensive summaries. The model outperforms baselines on ROUGE scores.

**Questions For The Authors:**

Can you provide more details on the human evaluations conducted in the experiments? How reliable are these evaluations, and are there any potential biases or limitations in the evaluation methodology?

**Reasons To Accept:**

1. The paper proposes a new model for Topic-Informed Dialogue Summarization that considers all topics present in a dialogue, reflecting dialogue topic distribution. This model is sound.
2. The model outperforms baselines on ROUGE scores.

**Reasons To Reject:**

The paper does not discuss the scalability of the proposed model, which could be a potential limitation in real-world scenarios. A discussion of the model's scalability and potential challenges in scaling it up could help readers better understand the practical implications of the proposed model.

**Reproducibility:**

3: Could reproduce the results with some difficulty. The settings of parameters are underspecified or subjectively determined; the training/evaluation data are not widely available.

**Reviewer Confidence:**

2: Willing to defend my evaluation, but it is fairly likely that I missed some details, didn't understand some central points, or can't be sure about the novelty of the work.

---

> ### Author Rebuttal · Authors · 2023-08-29
>
> > **Q1.** The paper does not discuss the scalability of the proposed model, which could be a potential limitation in real-world scenarios. A discussion of the model's scalability and potential challenges in scaling it up could help readers better understand the practical implications of the proposed model.
>
> **A1.** In real-world scenarios, the proposed dialogue summarization model can be applied not only to simple dialogues between two speakers, but also to multi-party dialogues, discussion summarization, etc. with more speakers and various topics. Appendix A.4 shows that the performance difference between TIDSum and BART-large is much larger in multi-topic than in single-topic. This suggests that it is applicable to dialogues with more diverse topics.
>
> > **Q2.** Can you provide more details on the human evaluations conducted in the experiments? How reliable are these evaluations, and are there any potential biases or limitations in the evaluation methodology?
>
> **A2.** Human evaluations were conducted by three linguistic experts. For both the golden summary and the output summary of all models, the evaluators were blinded to which model produced the output. There are no potential biases or limitations because the evaluators assessed the randomly mixed summaries without knowing which model the output was for. To assess inter-annotator agreement, we measured the Fleiss' kappa coefficient and achieved scores of 0.43, 0.35, and 0.40 for informativeness, conciseness, and coverage, respectively.

---

### Official Review · Reviewer_KN1n · 2023-08-04

**Typos Grammar Style And Presentation Improvements:** 1. For the human evaluation part, can…
**Soundness:** 3

**Excitement:**

3: Ambivalent: It has merits (e.g., it reports state-of-the-art results, the idea is nice), but there are key weaknesses (e.g., it describes incremental work), and it can significantly benefit from another round of revision. However, I won't object to accepting it if my co-reviewers champion it.

**Paper Topic And Main Contributions:**

The paper proposes a new dialogue summarization model that minimizes the gap between summary and dialogue topic distribution. And the model achieves the SOTA performance in SAMSum and DialogSum datasets.

**Questions For The Authors:**

A. I am wondering how you train the topic extractor.  Since latent topic embedding lte is derived from the TopClus and dti is the mean pooling of all tokens, they are projected in different semantic embedding spaces. Do you have one extra step besides summary training for the topic extractor?

**Reasons To Accept:**

1. The paper is well written and the model achieves the SOTA performance in SAMSum and DialogSum datasets.


**Reasons To Reject:**

1. The paper selected "number of topics to 5 for SAMSum and 7 for DialogSum". It seems the dataset contains very limited topics which is an uncommon scenario. Given the relatively short length of these two datasets, one may question the effectiveness of the proposed method when applied to dialogues with a more extensive range of topics and longer length.
2. while dialogue contains multiple topics, the summary of dialogue doesn't necessarily need to cover all the topics mentioned in the dialogue. Can you test the topic similarity of the gold standard summary and dialogue inputs to see what's the upper bound?

**Reproducibility:**

3: Could reproduce the results with some difficulty. The settings of parameters are underspecified or subjectively determined; the training/evaluation data are not widely available.

**Reviewer Confidence:**

3: Pretty sure, but there's a chance I missed something. Although I have a good feel for this area in general, I did not carefully check the paper's details, e.g., the math, experimental design, or novelty.

---

> ### Author Rebuttal · Authors · 2023-08-29
>
> > **Q1.** The paper selected "number of topics to 5 for SAMSum and 7 for DialogSum". It seems the dataset contains very limited topics which is an uncommon scenario. Given the relatively short length of these two datasets, one may question the effectiveness of the proposed method when applied to dialogues with a more extensive range of topics and longer length.
> > **Q2.** While dialogue contains multiple topics, the summary of dialogue doesn't necessarily need to cover all the topics mentioned in the dialogue. Can you test the topic similarity of the gold standard summary and dialogue inputs to see what's the upper bound?
>
> **A1 & A2.** We'll write the responses to Q1 and Q2 together. We have merely found the optimal number of topics that can appear in the entire dataset to get the highest performance. If a wider range of topics and longer dialogues are to be covered, a larger number of topics can be specified and trained on. Note that setting the number of topics to five does not ensure that five topics will appear, but rather that this is the maximum number of topics that can appear. We sampled 50 examples from the SAMSum dataset and counted the number of topics with probability ≥ 0.1. The dialogue topic distribution was dominated by about 2.1 topics, and the golden summary topic distribution was dominated by about 1.7 topics. Table 1 is an example of the specific distributions. Even if a larger dataset is used and 11 topics are designated, there will still be dominant topics in any given dialogue. We agree that you don't need to cover every topic in the dialogue to generate the summary. However, our approach allows the summary to better reflect a few of the more influential topics in the dialogue.
>
> |Index|Dialogue Topic Distribution|Summary Topic Distribution|
> |:---:|---|---|
> |1|[0.0004, 0.0010, 0.0003, 0.2984, 0.6999]|[0.0002, 0.0016, 0.0003, 0.4693, 0.5285]|
> |2|[0.0018, 0.1442, 0.0014, 0.4161, 0.4365]|[0.0009, 0.1376, 0.0009, 0.0856, 0.7750]|
> |3|[0.6036, 0.0012, 0.2248, 0.0011, 0.1693]|[0.6178, 0.0002, 0.3782, 0.0001, 0.0037]|
> |4|[0.0018, 0.9308, 0.0001, 0.0671, 0.0002]|[0.0007, 0.9143, 0.0012, 0.0002, 0.0836]|
>
> **Table 1. Samples of the dialogue topic distribution and the golden summary topic distribution in the SAMSum dataset**
>
>
> > **Q3.** I am wondering how you train the topic extractor. Since latent topic embedding $lte$ is derived from the TopClus and $dti$ is the mean pooling of all tokens, they are projected in different semantic embedding spaces. Do you have one extra step besides summary training for the topic extractor?
>
> **A3.** $lte$ and $dti$ are not in different semantic embedding spaces. Two $lte$ concatenate to become a $tip$, which gives topic information to other tokens through attention, and also influences the generation of $dti$. The dialogue topic distribution $p(t_k|lte)$ extracted from TopClus is frozen distribution because it has been extracted previously. A single MLP, topic extractor, extracts the summary topic distribution from $dti$ and is trained to make the summary topic distribution similar to the pre-extracted dialogue topic distribution with cross entropy loss. In other words, topic extractor is trained to make the summary topic distribution equal to the dialogue topic distribution. As the topic extractor is trained to get better at extracting the summary topic distribution, the decoder is trained to generate summaries better.
>
>
> > **Q4.** For the human evaluation part, can you give a formula definition of what's conciseness and coverage?
>
> **A4.** The metrics used for human evaluation just followed the approach of _Feng et al. 2021b_. Among these metrics, conciseness is a metric that measures the extent to which the summary discards redundant information. The number of incorrectly generated sentences or unnecessary sentences $k$ for all $l$ sentences is calculated by the following formula:
> $$Round(5*(1-\frac{k}{l}))$$
>
> Coverage is a measure of how well the summary reflects all the key topics that appeared in the dialogue. If there are $n$ topics of dialogue that the evaluators think should be included in the summary, and the generated summary contains $m$ $(m≤n)$ of them, the formula is as follows:
> $$Round(5*\frac{m}{n})$$

---

### Official Review · Reviewer_GoNj · 2023-08-04

**Soundness:** 3

**Excitement:**

3: Ambivalent: It has merits (e.g., it reports state-of-the-art results, the idea is nice), but there are key weaknesses (e.g., it describes incremental work), and it can significantly benefit from another round of revision. However, I won't object to accepting it if my co-reviewers champion it.

**Paper Topic And Main Contributions:**

Since the context of dialogues with multiple topics prone to topic drift, the authors propose a novel dialogue summarization model called Topic-Informed Dialogue Summarizer (TIDSum), which aims to capture and reflect all the topics present in dialogues. They utilize an effective topic discovery model, called TopClus, to estimate the distribution of multiple topics within a dialogue. Then they conduct a task-specific soft-prompt called the "topic-informed prompt", which is derived from the latent embedding of the dialogue obtained from the TopClus model. Finally, by employing the topic-informed prompt, the model influences the context vectors of the encoder and decoder, ensuring that the topic information is considered throughout both encoding and decoding phases of summarization.

**Reasons To Accept:**

1)The proposed Topic-Informed Dialogue Summarizer (TIDSum) explicitly considers the distribution of multiple topics within a dialogue. This approach allows the generated summaries to comprehensively capture all the important topics present in the dialogue, resulting in summaries that are more comprehensive and representative of the original conversation.
2)The paper introduces the concept of a "topic-informed prompt" that is integrated into the summarization model's encoding and decoding processes. This prompt is derived from the latent embedding of the TopClus model, which estimates the distribution of dialogue topics. By leveraging this prompt, the model enhances the context vectors of both the encoder and decoder, ensuring that topic information is effectively captured throughout the summarization process.
3)The experimental results presented in the paper indicate that the proposed TIDSum model outperforms existing methods on two daily dialogue summarization datasets, SAMSum and DialogSum.

**Reasons To Reject:**

1）The author's model description is not clear. For example, in line 149, I don't quite understand how the tip is calculated, please explain further here or use a formula example to illustrate
2）Insufficient experiments. Only ablation experiments and main experiments are shown, lack of model analysis experiments and sample analysis.

**Reproducibility:**

3: Could reproduce the results with some difficulty. The settings of parameters are underspecified or subjectively determined; the training/evaluation data are not widely available.

**Reviewer Confidence:**

4: Quite sure. I tried to check the important points carefully. It's unlikely, though conceivable, that I missed something that should affect my ratings.

---

> ### Author Rebuttal · Authors · 2023-08-29
>
> > **Q1.** The author's model description is not clear. For example, in line 149, I don't quite understand how the $tip$ is calculated, please explain further here or use a formula example to illustrate.
>
> **A1.** $tip$ (1024 dims) is a concatenation of two $lte$ (512 dims). Since $lte$ is a hidden state of the TopClus autoencoder, it must be smaller than TopClus autoencoder input dimension (768 dims). The TopClus input is fixed at 768 dimensions because it origins from the CLS token of BERT that encodes the input dialogue. Thus, we set it to 512 dimensions to easily match the input dimensions of BART-large (1024 dims) by concatenating two $lte$ (1024 dimensions) as follows:
>
> $$tip = lte \oplus lte$$
>
> where $\oplus$ means concatenation.
>
> > **Q2.** Insufficient experiments. Only ablation experiments and main experiments are shown, lack of model analysis experiments and sample analysis.
>
> **A2.** We conducted several additional analytical experiments, but for the limited space of the short paper, only the core experiments were presented in the main text. Appendix A.4 shows an evidence that our proposed model performs better in summarizing dialogues with multiple topics. When we evaluated the generation results by dividing the dialogues into single-topic and multi-topic dialogues, the performance difference between BART-large (baseline) and TIDSum was much larger for multi-topic dialogues. It is an important evidence that our model works better in complex dialogues with multiple topics (e.g., multi-party dialogues) than in relatively simple and short dialogues with a single topic.
>
> As shown in Table 1, we also found that the average attention weight of topic-informed prompt ($tip$) are relatively higher than the other tokens, which means that the topic information in $tip$ is more concentrated in the sequence, allowing the model to generate more meaningful summaries.
>
> |Average attention weight of $tip$|Average attention weight of the other tokens|
> |:---:|:---:|
> |**0.3044**|0.0197|
>
> **Table 1. Comparison of the average attention weight of $tip$ to the average attention weight of the other tokens, for 50 examples**

---

### Meta-Review · Area_Chair_PyGq · 2023-09-11

**Recommendation:** 3

**Metareview:**

The paper introduces topic-informed dialogue summarisation, which pre-computes the topic distribution in a dialogue and incorporates this information (in the form of soft prompt / latent embeddings) in the summarisation process. Reviewers thought the idea of incorporating topic distribution, particularly for dialogues, is well-motivated and the proposed approach is interesting. Moreover, reviewers also thought quantitatively the topic-informed summariser appears to perform very well. That said, there are some minor but important concerns that dampens the excitement of reviewers: (1) the writing for certain parts in the method description can be improved (GoNj and KN1n); (2) the paper would benefit from more analyses to bring out more insights (GoNj); (3) the utility/applicability of the approach for longer conversations warrants at least a discussion, if not more results (KN1n, Fkky); and (4) the paper might want to provide some results to validate the assumption that the topic distribution in the ground truth summary is similar to that of the dialogue.

---

### Decision · Program_Chairs · 2023-10-07

**Decision:**

Accept-Findings

**Comment:**

The paper introduces topic-informed dialogue summarisation, which pre-computes the topic distribution in a dialogue and incorporates this information (in the form of soft prompt / latent embeddings) in the summarisation process. Reviewers thought the idea of incorporating topic distribution, particularly for dialogues, is well-motivated and the proposed approach is interesting. Moreover, reviewers also thought quantitatively the topic-informed summariser appears to perform very well. That said, there are some minor but important concerns that dampens the excitement of reviewers: (1) the writing for certain parts in the method description can be improved (GoNj and KN1n); (2) the paper would benefit from more analyses to bring out more insights (GoNj); (3) the utility/applicability of the approach for longer conversations warrants at least a discussion, if not more results (KN1n, Fkky); and (4) the paper might want to provide some results to validate the assumption that the topic distribution in the ground truth summary is similar to that of the dialogue.